# Renal Function in Chronic Hepatitis C Patients in Mongolia

**DOI:** 10.3390/diagnostics15121471

**Published:** 2025-06-10

**Authors:** Gantogtokh Dashjamts, Amin-Erdene Ganzorig, Yumchinsuren Tsedendorj, Ankhzaya Batsaikhan, Dolgion Daramjav, Enkhmend Khayankhyarvaa, Bolor Ulziitsogt, Otgongerel Nergui, Nomin-Erdene Davaasuren, Ganchimeg Dondov, Tegshjargal Badamjav, Tulgaa Lonjid, Chung-Feng Huang, Tzu-Chun Lin, Batbold Batsaikhan, Chia-Yen Dai

**Affiliations:** 1Department of Internal Medicine, Institute of Medical Sciences, Ministry of Economy and Development, Ulaanbaatar 14210, Mongolia; d.gantogtoh.ims@gmail.com (G.D.); aminerdene.ims@mnums.edu.mn (A.-E.G.); yumchinsuren.ims@mnums.edu.mn (Y.T.); ankhzaya.ims@mnums.edu.mn (A.B.); dolgion.ims@mnums.edu.mn (D.D.); enkhmend.ims@mnums.edu.mn (E.K.); bolor.ims@mnums.edu.mn (B.U.); otgongerel.ims@mnums.edu.mn (O.N.); nominerdene.ims@mnums.edu.mn (N.-E.D.); ganchimeg.ims@mnums.edu.mn (G.D.); tegshjargal.ims@mnums.edu.mn (T.B.); tulgaa.ims@mnums.edu.mn (T.L.); 2Department of Biological Sciences, School of Life Sciences, Inner Mongolia University, Hohhot 010031, China; 3Hepatobiliary Division, Department of Internal Medicine, Department of Occupational and Environmental Medicine, Kaohsiung Medical University Hospital, Kaohsiung Medical University, Kaohsiung 807378, Taiwan; huangcf@kmu.edu.tw (C.-F.H.); tclin1983@gmail.com (T.-C.L.); 4College of Medicine, Drug Development and Value Creation Research Center, Kaohsiung Medical University, Kaohsiung 807378, Taiwan; 5Department of Health Research, Graduate School, Mongolian National University of Medical Sciences, Ulaanbaatar 14210, Mongolia; 6Department of Biological Science and Technology, College of Biological Science and Technology, National Yang Ming Chiao Tung University, Hsinchu 30044, Taiwan; 7College of Professional Studies, National Pingtung University of Science and Technology, Pingtung 91201, Taiwan

**Keywords:** hepatitis C virus infection, glomerular filtration rate, liver fibrosis, extrahepatic manifestation, chronic kidney disease

## Abstract

**Background:** According to a study conducted among a relatively healthy population of Mongolia (2017), the prevalence of hepatitis C virus (HCV) infection is 8.5%, which is considered a high prevalence of this infection. In addition to inflammation of the liver, other organ systems are affected by HCV infection, according to research. Our study aimed to evaluate renal dysfunction in patients with HCV infection. **Methods:** In the study, 111 people with chronic hepatitis C virus infection were included in the study group, and 111 relatively healthy people were included in the control group. Laboratory parameters were analyzed. Liver fibrosis score was assessed and evaluated by renal function. **Results:** There were 22.9% (51) men and 77.1% (171) women among the 222 participants, and the average age was 40.7 ± 11.1 years. The glomerular filtration rate was 105.3 ± 24.5 in the chronic hepatitis C virus-infected group and 118.7 ± 18.5 in the control group, or the statistically significant difference in the case group compared to the control group was *p* < 0.01. The liver fibrosis score was higher in the case group than in the control group. According to logistic regression analysis, patients with hepatitis C virus infection are 25 times more likely to have a decrease in glomerular filtration rate than those without viral infection (OR 24.91, 95% CI 3.13–198.38, *p* = 0.002). **Conclusions:** Our study showed that HCV infection leads to kidney function loss. In addition, older age, obesity, and severe liver fibrosis contribute to kidney function decline.

## 1. Introduction

According to the World Health Organization, approximately 71 million people are infected with the hepatitis C virus (HCV) globally, and in 2019, more than 299,000 people died from hepatic cirrhosis and hepatocellular carcinoma (HCC) due to HCV infection [1]. In a study conducted among a relatively healthy population in Mongolia, the prevalence of HCV infection was found to be 8.5%, classifying the country as having a high prevalence of the viral infection [2]. Studies indicate that in addition to causing hepatitis, HCV infection can also lead to pathological changes in other organ systems, especially the kidneys [3]. Among the renal disorders caused by HCV infection, the most commonly reported are membranoproliferative glomerulonephritis (MPGN) and membranous nephropathy [4]. While proteinuria has been identified as a symptom arising from HCV infection, findings regarding its impact on glomerular filtration rate remain inconsistent. Variations in study results may be attributed to differences in HCV genotype, viral load, and other factors [5].

The prevalence of chronic kidney disease (CKD) is increasing, with a rapid rise in cases of end-stage renal disease (ESRD) annually not only in Mongolia but globally, making it a major global public health problem. The disease is diagnosed either by a decrease in the glomerular filtration rate (GFR) or the presence of albumin in the urine [6]. GFR is an indicator of renal function and is defined as the volume of blood filtered through the kidneys per minute [7]. Estimation of GFR is not only a standard method for evaluating CKD based on diagnostic classification criteria but also determines the five stages of kidney disease, regardless of the level of proteinuria [8].

According to a study by Li et al. the GFR of individuals with HCV infection was lower than that of uninfected individuals (88–102 mL/min/1.73 m^2^, *p* < 0.001), with significantly lower GFR observed across all stages of CKD [9]. In a study by Hsu et al., the majority of HCV-infected patients tended to be male and had comorbidities such as diabetes mellitus and arterial hypertension [10]. In a study conducted in Taiwan, the HCV-infected group showed a significantly decreased GFR compared to the control group. Furthermore, HCV-infected individuals had a higher likelihood of developing metabolic syndrome, and conditions such as obesity, arterial hypertension, and diabetes mellitus were found to affect renal function negatively. Moreover, the comorbidity of HCV infection with arterial hypertension and obesity significantly increased the risk of GFR reduction [11].

The number of individuals with CKD was found to be higher in the HCV-infected group compared to those without viral infections [12]. Although significant progress has been made globally and in Mongolia in the regard of treatment of viral hepatitis, authors continue to report that the diagnosis and treatment of extrahepatic manifestations of HCV, particularly renal involvement, remain inadequate. Thus, the current study aims to investigate renal function impairment among individuals infected with HCV, a virus that has a high prevalence among the Mongolian population.

## 2. Materials and Methods

### 2.1. Study Design, Scope, and Sample

The descriptive case-control study was conducted from June 2021 to March 2023 at the State Third Central Hospital (STCH) and the National Center for Communicable Diseases (NCCD) in Ulaanbaatar, Mongolia. The case group consisted of 111 patients diagnosed with hepatitis C virus (HCV) infection who were hospitalized at the Adult Hepatology Unit of NCCD and the Department of Gastroenterology at STCH. The control group comprised 111 relatively healthy individuals who participated in routine health screening programs at STCH, General Health Center for Songinokhairkhan District, and General Health Center for Bayanzurkh District. Cases and controls were matched by age (±1 year) and sex and the total number of participants was 222. Inclusion criteria for the case group were (a) confirmed HCV infection; (b) no co-infection with other hepatitis viruses; and (c) age ≥ 18 years. Inclusion criteria for the control group were (a) no HCV or other hepatitis virus infections; and (b) matched by age and sex with the case group. Exclusion criteria for both groups were (a) incomplete laboratory or diagnostic data; (b) pregnant; (c) diagnosed with chronic kidney disease (CKD); (d) presence of other comorbid conditions known to impair renal function (e.g., systemic or hereditary diseases); (e) diagnosed hepatic or other organ malignancies; (f) evident metabolic disorders; (g) history of antiviral treatment for HCV infection; (h) have co-infection of hepatitis B, hepatitis D, or human immunodeficiency virus; and (i) diagnosed with arterial hypertension or diabetes mellitus. 

### 2.2. Questionnaire, Laboratory Tests, and Physical Exam

Data Collection and Measurements: Data on participants’ general demographics, hepatitis C virus (HCV) infection status, medication use, and renal risk factors was collected using a structured questionnaire.

Blood pressure: Blood pressure was measured on both arms using a sphygmomanometer while participants were seated and at rest.

Diabetes assessment: Diagnosed as having diabetes if fasting glucose ≥ 7.0 mmol/L, HbA1C ≥ 6.5%, or if taking diabetes medications or insulin injections/Mongolian Clinical guidelines for type 2 diabetes, 2021 Order No. A196/. Patients diagnosed with diabetes were excluded from the study. Participants with fasting glucose of 6.1–6.9 mmol/L or changes in fasting glucose were included in the study.

Body Mass Index (BMI): BMI was calculated by dividing body weight (in kilograms) by the square of height (in meters). The classification of obesity was based on the criteria established by the World Health Organization (WHO).

Laboratory tests:

Serum HBsAg and anti-HCV were detected using serological tests at the laboratories of the determined. Laboratory indicators listed in the study form were tested at the central laboratories of the State Third Central Hospital (STCH), the National Center for Communicable Diseases (NCCD), General Health Center for Songinokhairkhan, and General Health Center for Bayanzurkh District. Results were recorded using a standardized data collection form and subsequently summarized.

Hepatic steatosis (fatty liver disease) was assessed by radiologists.

Diagnostic criteria for fatty liver disease:Mild—slightly increased echogenicity of the liver, loss of intrahepatic arterial boundaries, normal appearance of the diaphragm.Moderate—markedly increased echogenicity of the liver, loss of visibility of the lower hepatic mass, and lack of clear visibility of the diaphragm.Severe—markedly increased echogenicity of the liver, with no clear differentiation of the diaphragm. Clinical guidelines for the diagnosis and treatment of non-alcoholic fatty liver disease, Order of the Ministry of Health of Mongolia, No. A/697, 2024.

For the interpretation of laboratory test results, units of measurement were standardized using the online calculator available at www.mdcalc.com/calc/76/mdrd-gfr-equation (accessed on 18 April 2025) to estimate glomerular filtration rate (GFR), creatinine clearance, and hepatic fibrosis scores.

Lipid profile: Total cholesterol, high-density lipoprotein (HDL), low-density lipoprotein (LDL), and triglycerides were measured by biochemistry. Lipid profile indicators were evaluated in accordance with the guidelines provided by the World Health Organization (WHO), the Ministry of Health of Mongolia, and the methodology used in the Millennium Challenge Corporation’s survey on the prevalence and risk factors of non-communicable diseases, injuries, and accidents.

### 2.3. The Method to Evaluate the Hepatic Fibrosis Degree

In the practice of hepatology, non-invasive methods such as APRI and FIB-4 scores are commonly used to assess hepatic fibrosis.

#### 2.3.1. Liver Fibrosis Grade Was Assessed Using the Fibrosis-4 (FIB4), APRI Index [13], Which Was Calculated Using the Following Formula


FIB4=(age [years]×AST [U/L])/(Platelet [109/L]×√ALT [U/L])


Table 1 presents the evaluation of liver fibrosis stage using the FIB-4 method (Table 1).

#### 2.3.2. Aspartate Aminotransferase to Platelet Ratio Index (APRI) Was Calculated Using Formula [13]


APRI=(GOT [U/L]/GOT [Upper limit of normal range])/Platelet [109/L]



$$APRI=\frac{\begin{array}{c} \mathrm{AST}\;\mathrm{level}\\ \mathrm{ULN}* \end{array}}{Platelet\;counts\;(10^{9}/L)}\times 100$$


### 2.4. Evaluation of Renal Function

#### 2.4.1. The Method to Evaluate the Glomerular Filtration Rate

The glomerular filtration rate was calculated using the MDRD GFR Equation [14].


$$\mathrm{eGFR}\;(\mathrm{mL}/\min)\;1.73\;m^{2}=186\times \mathrm{Cr}\;(\mathrm{mg}/\mathrm{dL})-1.154\times \mathrm{age}\;(\mathrm{year})-0.203\times 0.74\;(\mathrm{female})$$


Normal kidney function: eGFR above 90 mL/min/1.73 m^2^ and no proteinuria;CKD stage 1: eGFR below 90 mL/min/1.73 m^2^ with evidence of kidney damage;CKD stage 2 (mild): eGFR of 60 to 89 mL/min/1.73 m^2^ with evidence of kidney damage;CKD stage 3 (moderate): eGFR of 30 to 59 mL/min/1.73 m^2^;CKD stage 4 (severe): eGFR of 15 to 29 mL/min/1.73 m^2^;CKD stage 5 kidney failure: eGFR less than 15 mL/min/1.73 m^2^.

Mild chronic kidney disease corresponds to stages I–II, while moderate to severe forms include stages III–IV. Chronic kidney disease is classified into five stages based on the glomerular filtration rate (GFR) [14].

#### 2.4.2. Method for Calculating Creatinine Clearance [15]


(1)
$$CrCl\;(mL/min)=\frac{\left(140-age\right)\times Lean\;Body\;Weight\left(kg\right)}{Serum\;Creatinine\left(\frac{mg}{dL}\right)\times 72}\times (0.85\;if\;female)$$


“Ach” Medical University Ethics Committee held a meeting on 30 June 2022 and received ethical approval in accordance with Resolution No. 222/05. The names of the participants and personal information were entered into a Microsoft Excel file, numerically coded, and confidentiality was strictly maintained. The study findings were reviewed by the Ethics Subcommittee of the Institute of Medical Sciences, Mongolia, and the expert evaluation concluded that there were no ethical violations or misconduct (Approval No. 2025/01).

### 2.5. Statistical Analysis

We computed mean values and standard deviations for continuous variables. In descriptive statistical analyses, continuous variables were reported as mean ± standard deviation (if the data follow a normal distribution) and median (minimum–maximum, if the data do not follow a normal distribution). Categorical variables were presented as numbers (count of occurrences) and percentages (%-relative frequency). Continuous and categorical variables were compared using Chi-square (X2), or Fisher’s exact test, either parametric or nonparametric statistical tests were employed. Continuous variables are summarized as mean ± standard deviation and were compared using Student’s *t*-test. To evaluate the relationship between associated factors and renal function, we conducted both univariate analysis and multivariate logistic regression analysis. Patient data were originally collected and organized using Microsoft Excel software (version 2016). Statistical significance was defined at a *p*-value below 0.05, considering two-sided hypotheses. All statistical computations were performed using IBM SPSS Statistics for Windows, Version 26.0, developed by IBM Corp. in Armonk, NY, USA.

## 3. Results

### 3.1. General Demographics of the Participants

A total of 222 individuals aged between 21 and 77 years were included in the study, of whom 22.9% (n = 51) were male and 77.1% (n = 171) were female. The mean age of individuals was 40 years, and all individuals in both groups were matched by age and sex (mean age: 40 ± 11.1 years). Of the 222 participants, renal function was assessed in 72.9% (n = 162) based on serum creatinine levels measured by laboratory testing.

In the group with hepatitis C virus (HCV) infection, glomerular filtration rate (GFR) and creatinine clearance were significantly lower compared to the control group (*p* < 0.01). HCV infection was associated with the following factors: prolonged prothrombin time (13.4 ± 2.2 vs. 9.5 ± 5.3; *p* < 0.001), elevated bilirubin levels (94.1 ± 86.2 vs. 14.06 ± 34.6; *p* < 0.001), increased AST levels (500.6 ± 517.0 vs. 41.7 ± 55.0; *p* < 0.001), increased ALT levels (764.4 ± 724.6 vs. 39.6 ± 46.1; *p* < 0.001), and decreased urea levels (3.2 ± 1.7 vs. 6.4 ± 6.8; *p* < 0.01). The FIB-4 and APRI scores, which estimate hepatic fibrosis, were higher in the case group compared to the control group. The number of participants with fasting glucose variability (FGV) was higher in the control group compared to the case group (Table 2).

### 3.2. The Comparison Between Study Variables

Among the participants, 95 (59.4%) had a glomerular filtration rate (GFR) below 120, while 65 (40.6%) had a GFR above 120. The average age in the group with GFR below 120 was 54.30 (±8.9) years, while in the group with GFR above 120, the average age was 52.30 (±12.0) years, indicating a statistically significant difference (*p* < 0.01). Creatinine clearance was significantly lower in the group with GFR below 120. However, the percentage of participants with hepatitis C virus infection was higher in the group with GFR below 120 (69%). The number of participants with FGV was higher in the group with a GFR below 120 mL/min/1.73 m^2^ (Table 3).

The average age of participants with a glomerular filtration rate (GFR) below 120 was 46.0 in the case group and 44.1 in the control group. In contrast, the average age of participants with a GFR above 120 was 35.2 in the case group and 34.9 in the control group. This indicates that as participants’ age increases, the glomerular filtration rate tends to decrease. Creatinine clearance was lower in individuals with a GFR below 120. Moreover, individuals who are overweight (BMI ≥ 25) showed a GFR below 120. However, no significant differences in hepatic fibrosis scores were observed between the groups. In both the case and control groups, the decrease in GFR below 120 was statistically significantly associated with age and being overweight (BMI ≥ 25) (*p* < 0.01). The number of participants with FGV was higher in the group with HCV infection and a GFR of less than 120. However, among participants without HCV infection, FGV was more common in the group with a GFR of less than 120. Having diabetes during HCV infection increases the risk of kidney function decline (Table 4).

On an evaluation of renal function in participants over 45 years of age in the case-control groups, the glomerular filtration rate (GFR) in the case group was 99.69 (±23.8), while in the control group, it was 111.05 (±16.7), indicating a statistically significant difference between the two groups (*p* < 0.01). However, creatinine clearance in the case group was 1.59 (±0.7) and in the control group, it was 1.81 (±1.7), with no significant difference between the two groups. In individuals over 45 years old, age had an effect on the reduction of glomerular filtration rate, but it was more pronounced in the case group with hepatitis C virus infection. When evaluating renal function in participants under 45 years of age in the case-control groups, the glomerular filtration rate in the case group was 111.52 (±22.6) and in the control group, it was 122.88 (±18.0). Creatinine clearance in the case group was 1.50 (±0.4) and in the control group, it was 1.91 (±1.1), with the hepatitis C virus-infected group showing significantly reduced renal function. A higher hepatic fibrosis score was observed in the case group for participants over 45 years of age (Table 5).

The figure shows that the GFR was significantly reduced in the group over 45 years old and infected with hepatitis C. However, the GFR was similar in the group over 45 years old without hepatitis C and the group under 45 years old with hepatitis C (Figure 1A). There is a statistical difference in the GFR of the group with a body mass index (BMI) of more than 25 and with HCV, which is 100.56, and the control group is 114.6, while the GFR of the group with a BMI of less than 25 and with HCV, which is 114.22, and the control group is 124.38 (*p* < 0.001). This shows that having HCV and being overweight has a greater impact on the decrease in GFR (Figure 1B). The eGFR rate of the HCV-infected and FIB4 > 3.25 group was 100.39, and the eGFR of the FIB4 < 3.25 group was 107.46, indicating that advanced fibrosis during HCV infection may impair renal function. However, the eGFR of the control group and FIB4 > 3.25 group was 128.75, and the eGFR of the FIB4 < 3.25 group was 116.76, which was statistically different (Figure 1C). Co-infection with HCV in people with FGV increases the risk of kidney function loss (Figure 1D).

In the case group, when comparing the factors influencing renal function decline by hepatic fibrosis stage, it was found that individuals with a hepatic fibrosis score greater than 3.25 had an average age of 46.8 years, while those with a fibrosis score of 3.25 or lower had an average age of 37.9 years, with a statistically significant difference between the groups (*p* < 0.01). Furthermore, there were no statistical differences between the two groups in body mass index, renal function (eGFR, CrCl), APRI score, and FGV (Table 6).

### 3.3. The Factors Influencing the Reduction of Glomerular Filtration Rate

In the univariate logistic regression analysis, the following factors were significantly associated with decreased glomerular filtration rate (GFR): hepatitis C virus (HCV) infection (OR = 27.70; 95% CI: 3.58–214.13; *p* = 0.001), age > 45 years (OR = 3.55; 95% CI: 1.20–10.50; *p* = 0.022), body mass index (BMI) > 25 (OR = 3.65; 95% CI: 1.01–13.20; *p* = 0.048), and hepatic fibrosis score > 3.25 (OR = 2.89; 95% CI: 1.41–5.91; *p* = 0.004). In the multivariate logistic regression analysis, HCV infection remained 25 times related to decreased GFR, with an odds ratio of 24.91 (95% CI: 3.13–198.38; *p* = 0.002) (Table 7).

## 4. Discussion

In the current study, we evaluated renal dysfunction among individuals with HCV infection, which has a high prevalence in the Mongolian population. A total of 222 participants were included in the study, of whom 22.1% were male and 77.9% were female, with a mean age of 40.7 ± 11.1 years. Our findings indicated that in the HCV-infected group (case group), both the glomerular filtration rate (GFR) and creatinine clearance were significantly lower, and hepatic fibrosis scores were higher compared to the control group, with statistically significant differences (*p* < 0.001). These findings are consistent with a previous report that the prevalence of chronic kidney disease (CKD) was higher in HCV-infected individuals compared to healthy controls [12]. This is also consistent with previous meta-analyses conducted in Asia, which reported an odds ratio of 1.45 (95% CI: 1.27–1.65) for the association between HCV infection and CKD [16]. Lee et al. further demonstrated a link between HCV infection and advanced stages of CKD, with a 14.5% prevalence of HCV among patients with end-stage renal disease (ESRD) [17]. In a 2021 case–control study conducted in Taiwan involving 6720 individuals, the proportion of individuals with an eGFR < 60 mL/min/1.73 m^2^ was 41.8% in the HCV group and 21.8% in the non-HCV group [11]. In comparison, the current study found that 19.8% of the HCV group and 0% of the non-HCV group had decreased GFR, which may be attributable to the smaller sample size in our study (eGFR < 60 mL/min/1.73 m^2^). Several studies have hypothesized a potential association between HCV infection and the development of CKD. Multiple cohort and meta-analysis studies have confirmed that patients with HCV infection are at significantly increased risk for developing CKD [18] and progressing to ESRD compared to HCV-negative individuals [19]. In our study, cryoglobulinemia precipitate the most common extra-hepatic manifestation of HCV infection was positive in 6.7% of patients serum 3 months after the end of DAA treatment [20].

HCV impairs insulin metabolism, leading to insulin resistance and hyperinsulinemia, which accelerate renal cell proliferation and tissue damage [21]. Additionally, increased arterial stiffness in the kidneys may cause vascular ischemia, resulting in chronic impairment of renal function due to reduced blood supply [22]. In our study, the HCV-positive group showed lower total cholesterol and higher triglyceride levels compared to the control group. A study conducted in Taiwan reported [23] that after direct-acting antiviral (DAA) treatment, patients who achieved a sustained virologic response showed increased levels of triglycerides, total cholesterol, high-density lipoprotein (HDL), and low-density lipoprotein (LDL). In our study, it was not possible to assess lipid metabolism following HCV DAA treatment. Additionally, HCV infection can alter lipid metabolism even in the absence of other liver-damaging factors [23].

In a study by Martin P et al., among 167 HCV-infected patients with normal renal function, 26 (15.6%) experienced a 50% decline in eGFR, whereas no such decline was observed among 779 HCV-negative patients. This supports the hypothesis that HCV infection increases the risk of renal impairment in patients with previously normal renal function [24]. The progression of CKD in patients with HCV infection may also be influenced by comorbid conditions and underlying renal pathologies. In our study, the incidence rate of CKD associated with HCV was relatively low (217.3 per 1000 person-years) compared to other studies. This discrepancy could be explained by our small sample size, lack of proteinuria or urinary albumin–creatinine ratio assessment, or variations in viral genotypes. Thus, further studies with larger sample sizes and more detailed methodologies are essential. Although our findings support the potential role of HCV infection in increasing CKD risk and reducing eGFR [25], they are not entirely consistent with all previous studies [26]. Some cross-sectional studies [12] were conducted by Modification of Diet in Renal Disease (MDRD) or CKD-EPI equations to determine the CKD stage, consistent with our methodology. In the Taiwan study, a significant proportion of participants were female (57.6%), and the HCV group exhibited statistically significant differences in BMI, cholesterol, triglycerides, uric acid, AST, ALT, creatinine (*p* < 0.001), and other biomarkers compared to the non-HCV group [10]. In contrast to our findings, the study by Lee V et al. [9] reported no association between age and reduced eGFR in both HCV-infected and non-infected groups, while in our study, eGFR decreased with advancing age. In our study, the prevalence of CKD among HCV-infected participants was 19.8%, which is similar to the 18.2% reported by Lee J et al. [17]. Their study also observed a higher prevalence of early-stage CKD (stages 1 and 2) and a lower prevalence of advanced-stage CKD (stage ≥ 3), which is consistent with our findings. The relatively young average age (40.7 ± 11.1 years) of our participants and the use of different eGFR estimation methods may explain these discrepancies. Although definitive evidence directly linking HCV to renal dysfunction remains inconclusive, prior studies have shown that immune complexes containing HCV antigens, antibodies, and complement components can deposit in the glomeruli, leading to renal injury [27]. Some studies have also suggested that HCV infection may accelerate renal atherosclerosis [28,29]. Our findings provide some evidence that HCV may impair renal function. However, further studies are needed to elucidate the influence of age, sex, HCV viral activity, hepatic fibrosis stage, and metabolic syndrome on renal outcomes. Our observation that eGFR declined with age is consistent with a study [30] from Italy. This study found a higher proportion of overweight individuals in the HCV-infected group compared to controls. Similarly, in our study, univariate logistic regression analysis showed a statistically significant association between BMI > 25 and reduced eGFR. This could be explained by HCV-induced lipid metabolism disorders, such as dyslipidemia, which may lead to renal atherosclerosis, renal capillary ischemia, and ultimately impaired renal function. Therefore, further studies should explore the relationship between metabolic alterations, particularly, lipid metabolism, and renal outcomes in HCV infection. In our previous study, when assessing renal function after completion of DAA therapy in HCV-infected patients, GFR increased, and lipid and glucose metabolism improved in participants with CKD [31].

Our study investigated renal dysfunction and its associated factors among patients with hepatitis C virus (HCV) infection in Mongolia. There is limited research on this topic in our country. The findings highlight the importance of early detection of extrahepatic complications, such as renal impairment, in patients with chronic HCV infection before overt clinical manifestations appear.

According to a Korean study, it was found that individuals with HCV infection and low eGFR (<90 mL/min/1.73 m^2^) had a 32% higher risk of developing CKD compared to uninfected individuals [32]. A study in the Asian population reported equal gender distribution among HCV-infected individuals [33]. In contrast to our findings, Sui L et al. in Taiwan reported that HCV-infected males had a higher risk of ESRD than females [34]. This difference may be due to the low number of ESRD cases in our study population. A study by Molnar M.Z et al. in the United States found that HCV infection was a risk factor for stage 3 CKD (eGFR < 60 mL/min/1.73 m^2^) and doubled the risk of progression to ESRD [25]. In our study, AST, ALT, prothrombin time, creatinine, triglycerides, alkaline phosphatase, and bilirubin levels were higher in the HCV-infected group compared to the control group. Furthermore, LDL and cholesterol levels were lower, and triglyceride levels were higher which were consistent with findings from other studies. A meta-analysis by Fabrizi F et al. reported a statistically significant association between HCV infection and proteinuria, with HCV-infected individuals having a 29% higher risk of developing proteinuria [35]. It was impossible to assess proteinuria. However, several cross-sectional studies have shown that proteinuria is significantly more prevalent among HCV-infected individuals compared to those without HCV [36,37].

In our study, a comparison between the HCV-infected and non-infected groups revealed that, among laboratory parameters, the HCV-infected group had a prolonged prothrombin time and elevated levels of liver and biliary enzymes, including AST, ALT, total bilirubin, alkaline phosphatase, and gamma-glutamyl transferase (GGT). Differences in lipid profiles, specifically total cholesterol and triglycerides, were observed between the two groups, reflecting alterations in metabolic function. Interestingly, a higher proportion of participants with elevated fasting glucose levels was found in the HCV-negative group. This unexpected finding highlights the need for further investigation using more specific diagnostic tests for diabetes. The most critical indicators in our study, GFR and creatinine clearance, were more significantly reduced in the HCV-infected group. In addition, non-invasive assessment of liver fibrosis revealed that the mean fibrosis score was higher in the HCV-infected group compared to the control group. Based on several key differences identified between the case and control groups in our study, we recommend that regular kidney screening be performed in patients with HCV infection. This should include laboratory assessments of renal function, proteinuria, hematuria, blood pressure monitoring, and estimation of GFR. For patients without existing renal impairment, it is advisable to initiate DAA therapy and subsequently monitor GFR and creatinine clearance every six months to ensure early detection and management of any renal dysfunction.

Our study has several limitations. Firstly, the small number of HCV-infected patients in our study may have limited the assessment of factors such as age, fasting glucose abnormalities, and obesity. Second, since none of the participants showed albuminuria in their urine, relying solely on the glomerular filtration rate to evaluate renal dysfunction may be insufficient. Thirdly, we were unable to assess the impact of direct-acting antiviral treatment for HCV infection on renal function. Additionally, long-term monitoring of renal function in patients with HCV infection may help prevent immune-mediated glomerulonephritis.

## 5. Conclusions

The estimated glomerular filtration rate (eGFR) was significantly lower in the group with chronic HCV infection compared to the non-infected group (105.3 mL/min/1.73 m^2^ vs. 118.7 mL/min/1.73 m^2^; *p* = 0.001). Similarly, creatinine clearance was also significantly reduced in the chronic HCV-infected group (1.5 mL/min vs. 1.8 mL/min; *p* = 0.001). As the hepatic fibrosis scores estimated by FIB-4 and APRI increased, a corresponding decline in eGFR was observed. Factors such as HCV infection, advancing age, being overweight (BMI ≥ 25), and elevated hepatic fibrosis score (≥3.25) were all found to be associated with decreased glomerular filtration rate.

## Figures and Tables

**Figure 1 diagnostics-15-01471-f001:**
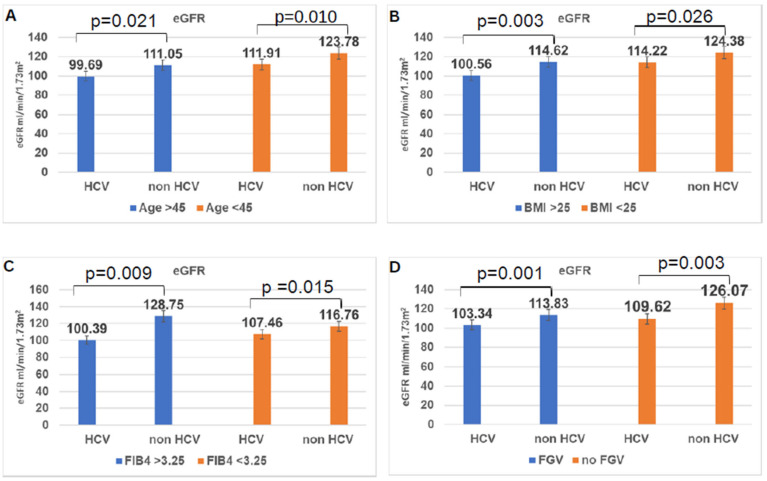
(**A**–**D**), Evaluation of glomerular filtration rate in HCV-infected and control groups stratified by age, diabetes, degree of liver fibrosis, and body mass index. eGFR = estimated glomerular filtration rate; HCV = Hepatitis C Virus; BMI = Body Mass Index; FIB4 = Index for liver fibrosis; FGV = Fasting Glucose Variability.

**Table 1 diagnostics-15-01471-t001:** Assessment of liver fibrosis (FIB4 method) [13].

Score	Fibrosis Degree	Diagnosis
<1.45	F0–F1	F0—normal F1—mild fibrosis F2–F3—moderate fibrosis F4–F6—severe fibrosis
1.45–3.25	F2–F3	
>3.25	F4–F6	

**Table 2 diagnostics-15-01471-t002:** The comparison of study variables between the two groups.

Characteristics	All (n = 222)	HCV (+) (n = 111)	HCV (−) (n = 111)	*p*-Value
Age (mean ± SD)	40.7 ± 11.1	40.7 ± 11.1	40.7 ± 11.2	0.962
Female, n (%)	171 (77.0%)	86 (50.3%)	85 (49.7%)	0.873
BMI (mean ± SD)	26.0 (4.8)	25.4 (4.0)	26.7 (5.4)	**0.045**
Platelet, mean (SD)	254.4 ± 69.3	257.6 ± 72.2	251.2 ± 66.6	0.490
PT (s)	11.3 ± 4.6	13.4 ± 2.2	9.5 ± 5.3	**<0.001**
GOT mean (SD)	275.4 ± 435.7	500.6 ± 517.0	41.7 ± 55.06	**<0.001**
GPT mean (SD)	410.4 ± 632.5	764.4 ± 724.6	39.6 ± 46.1	**<0.001**
Total bilirubin umol/L	65.2 ± 81.5	94.1 ± 86.2	14.06 ± 34.6	**<0.001**
ALP umol/L	185.3 ± 123.6	193.7 ± 124.4	90.4 ± 59.9	**0.016**
GGT U/L	223.9 ± 194.7	237.5 ± 194.7	36.4 ± 20.0	**0.045**
TCH mean (SD)	4.3 ± 1.01	4.1 ± 0.8	5.7 ± 0.9	**<0.001**
TG mean (SD)	1.5 ± 0.6	1.62 ± 0.8	1.45 ± 0.4	0.772
Creatinine mmol/L	56.2 ± 18.1	57.0 ± 17.6	55.6 ± 18.5	0.642
Urea mmol/L	4.9 ± 5.2	3.2 ± 1.7	6.4 ± 6.8	**<0.001**
FGV, N/n (%)	164/57	93/21 (22.5)	71/36 (50.7)	**<0.001**
eGFR mean (SD)	112.8 ± 22.3	105.3 ± 24.5	118.7 ± 18.5	**<0.001**
CrCl mean (SD)	1.7 ± 1.1	1.5 ± 0.6	1.8 ± 1.3	**<0.001**
FIB4 (SD)	2.4 ± 3.3	2.9 ± 2.9	1.9 ± 3.7	**<0.001**
APRI (SD)	3.0 ± 4.7	5.1 ± 5.7	0.8 ± 2.0	**<0.001**

We assessed renal function in 162 participants in the study. Because some participants had missing diagnostic data, it was not possible to assess renal function based on creatinine. Values are presented as mean (SD) or proportions. Bold values are statistically significant. Continuous and categorical variables were analyzed using Student’s *t*-test and the Chi-squared test to compare the HCV and control groups. HCV = Hepatitis C Virus; BMI = Body Mass Index; PT = Prothrombin Time; GOT = aspartate aminotransferase; GPT = alanine aminotransferase; ALP = alkaline phosphatase; GGT = gamma-glutamyl transferase; TCH = total cholesterol; TG = triglycerides; FGV = Fasting Glucose Variability; eGFR = estimated Glomerular Filtration Rate; CrCl = Creatinine Clearance; FIB4 = Index for liver fibrosis; APRI = AST to Platelet Ratio Index.

**Table 3 diagnostics-15-01471-t003:** Factors influencing renal function impairment among participants.

Characteristics	eGFR ≤ 120, n = 95	eGFR ≥ 120, n = 65	*p*-Value
Age (mean ± SD)	54.3 (8.9)	52.3 (12.0)	**0.048**
Male (%) Female (%)	28 (29.4%) 67 (70.5%)	13 (20.0%) 52 (80.0%)	0.178
BMI (mean ± SD)	24.4 (3.6)	24.8 (3.5)	0.276
CrCl mean (SD)	1.1 (0.2)	1.8 (1.1)	**<0.001**
FIB4 (SD)	3.5 (2.9)	3.1 (3.0)	0.136
APRI (SD)	1.9 (1.5)	1.8 (1.6)	0.352
Liver steatosis ^a^ (%)	58.4%	33.3%	0.216
Hyperlipidemia (%)	8.0%	14.2%	0.489
FGV (%)	73/34 (46.5)	49/15 (30.6)	0.078
HCV (+), (%)	49 (69.0%)	22 (31.0%)	**0.027**

We assessed renal function in 162 participants in the study. Because some participants had missing diagnostic data, it was not possible to assess renal function based on creatinine. Bold values are statistically significant. Continuous and categorical variables were analyzed using Student’s *t*-test and the Chi-squared test to compare the HCV and control groups. BMI = Body Mass Index; eGFR = estimated Glomerular Filtration Rate; CrCl = Creatinine Clearance; FIB4 = Index for liver fibrosis; APRI = AST to Platelet Ratio Index; FGV = Fasting Glucose Variability; HCV = Hepatitis C Virus; Liver steatosis ^a^ = Liver steatosis observed on abdominal ultrasound.

**Table 4 diagnostics-15-01471-t004:** The factors influencing glomerular filtration rate.

Parameter	HCV (+)			HCV (−)		
	eGFR ≤ 120	eGFR ≥ 120	*p*-Value	eGFR ≤ 120	eGFR ≥ 120	*p*-Value
Age (mean ± SD)	46.0 ± 10.0	35.2 ± 10.4	**<0.001**	44.1 ± 9.8	34.9 ± 10.3	**<0.001**
BMI (mean ± SD)	26.7 ± 3.4	23.6 ± 3.6	**<0.001**	27.8 ± 4.9	25.4 ± 5.9	**0.038**
CrCl mean (SD)	1.4 ± 0.5	1.7 ± 0.6	**0.048**	1.3 ± 0.2	2.4 ± 1.8	**<0.001**
FIB4 (SD)	3.1 ± 3.5	2.2 ± 1.7	0.263	1.9 ± 3.1	1.9 ± 4.7	0.996
APRI (SD)	5.7 ± 6.4	5.6 ± 4.9	0.959	1.1 ± 2.7	0.6 ± 1.2	0.252
FGV (%)	20/7 (35.0)	43/11 (25.5)	0.441	29/8 (27.5)	30/23 (76.6)	**<0.001**

We assessed renal function in 162 participants in the study. Because some participants had missing diagnostic data, it was not possible to assess renal function based on creatinine. Bold values are statistically significant. Continuous and categorical variables were analyzed using the *t* test and the Chi-squared test to compare the HCV and control groups. HCV = Hepatitis C Virus; eGFR = estimated Glomerular Filtration Rate; BMI = Body Mass Index; CrCl = creatinine clearance; FIB4 = Index for liver fibrosis; APRI = AST to Platelet Ratio Index; FGV = Fasting Glucose Variability.

**Table 5 diagnostics-15-01471-t005:** The comparison of variables between age groups.

Parameter	HCV (+)	HCV (−)	*p*-Value	HCV (+)	HCV (−)	*p*-Value
	Age > 45	Age > 45		Age < 45	Age < 45	
BMI (mean ± SD)	27.0 (3.4)	28.7 (5.4)	0.077	24.5 (4.1)	25.4 (4.9)	0.271
eGFR mean (SD)	99.6 (23.8)	111.0 (16.7)	**0.021**	111.5 (22.6)	122.8 (18.0)	**0.008**
CrCl mean (SD)	1.5 (0.7)	1.8 (1.7)	0.476	1.5 (0.4)	1.9 (1.1)	**0.014**
FIB4 (SD)	4.1 (3.8)	2.3 (3.5)	**0.020**	2.0 (1.6)	1.7 (3.7)	0.455
APRI (SD)	4.9 (5.3)	1.2 (2.8)	**<0.001**	5.0 (5.9)	0.5 (1.0)	**<0.001**

We assessed renal function in 162 participants in the study. Because some participants had missing diagnostic data, it was not possible to assess renal function based on creatinine. Bold values are statistically significant. Continuous and categorical variables were analyzed using Student’s *t*-test and the Chi-squared test to compare the HCV and control groups. HCV = Hepatitis C Virus; BMI = body mass index; eGFR = estimated Glomerular Filtration Rate; CrCl = Creatinine Clearance; FIB4 = Index for liver fibrosis; APRI = AST to Platelet Ratio Index.

**Table 6 diagnostics-15-01471-t006:** The comparison of factors influencing renal function decline by hepatic fibrosis stage in HCV (+) patients.

Characteristics	FIB4 ≥ 3.25	FIB4 ≤ 3.25	*p*-Value
Age (mean ± SD)	46.8 (9.3)	37.9 (10.8)	**<0.001**
BMI (mean ± SD)	26.2 (3.3)	25.0 (4.3)	0.139
eGFR mean (SD)	97.9 (28.8)	107.4 (22.6)	0.154
CrCl mean (SD)	1.4 (0.5)	1.5 (0.6)	0.530
APRI (SD)	5.6 (5.4)	4.7 (5.8)	0.496
FGV (%)	40/17 (42.5)	120/39 (32.5)	0.251

We assessed renal function in 162 participants in the study. Because some participants had missing diagnostic data, it was not possible to assess renal function based on creatinine. Bold values are statistically significant. Continuous and categorical variables were analyzed using Student’s *t*-test and the Chi-squared test to compare the HCV and control groups. HCV = Hepatitis C Virus; BMI = Body Mass Index; eGFR = estimated Glomerular Filtration Rate; CrCl = Creatinine Clearance; FIB4 = Index for liver fibrosis; APRI = AST to Platelet Ratio Index; FGV = Fasting Glucose Variability.

**Table 7 diagnostics-15-01471-t007:** The factors influencing glomerular filtration rate reduction.

	Univariate Logistic Regression			Multivariate Logistic Regression		
eGFR mL/min/1.73 m^2^	OR	95% CI	*p*-Value	OR	95% CI	*p*-Value
HCV (+)	27.70	3.58–214.13	<0.001	24.91	3.13–198.38	0.002
Age ≥ 45 (mean ± SD)	3.55	1.20–10.50	0.022	2.18	0.60–7.92	0.232
BMI ≥ 25 (mean ± SD)	3.65	1.01–13.20	0.048	2.63	0.61–11.3	0.193
FIB4 ≥ 3.25 (SD)	2.89	1.41–5.91	0.004	0.92	0.26–3.29	0.909
Gender (female)	0.88	0.29–2.64	0.824	1.20	0.35–4.16	0.764
FGV mmol/L	0.88	0.28–2.72	0.827	0.55	0.14–2.03	0.371

We assessed renal function in 162 participants in the study. Because some participants had missing diagnostic data, it was not possible to assess renal function based on creatinine. Adjusted odds ratio (OR) with 95% confidence intervals (CIs) and their *p*-values were calculated after adjusting for HCV, age, BMI, FIB4, gender, and diabetes using the log regression model. A multiple logistic regression model with forward selection was used to estimate the beta regression coefficient, ***p***-value, as well as odds ratios (ORs) and their 95% confidence intervals (CIs) for each of the selected risk predictors. The significant risk predictors were entered into this model (*p* < 0.005). HCV = Hepatitis C Virus; BMI = Body Mass Index; eGFR = estimated Glomerular Filtration Rate; CrCl = Creatinine Clearance; FIB4 = Index for liver fibrosis; FGV = Fasting Glucose Variability.

## Data Availability

Data are contained within the article.

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
