# Peer review of "Renal Function in Chronic Hepatitis C Patients in Mongolia"

_diagnostics, 2025, doi:10.3390/diagnostics15121471_

Round 1
Reviewer 1 Report
Comments and Suggestions for Authors
General review: This manuscript investigates the impact of chronic hepatitis C virus (HCV) infection on renal function in a Mongolian population, a region with a high HCV prevalence. The findings suggest that HCV infection is associated with reduced renal function, and this decline is further exacerbated by factors such as older age, obesity, and severe liver fibrosis.
Specific review:
- Introduction: please clarify what "viral activity" refers to when discussing inconsistencies in findings regarding GFR. (e.g., viral load, genotype).
- Material and methods:
- The exclusion criterion "h) Diagnosed with arterial hypertension or diabetes mellitus" appears to contradict the data in Table 2 and Table 4 where diabetes is present and analyzed as a variable, and is significantly different between cases/controls or GFR groups. This is a major inconsistency that needs to be addressed. If patients with pre-existing DM/HTN were not excluded, the criteria should be revised! If they were excluded, the data presentation needs to be explained (e.g., incident cases during study, or misclassification).
- How was "relatively healthy" for the control group ascertained beyond "no HCV or other hepatitis virus infections"? Were they screened for other conditions that could affect renal function?
- It states "Hepatic steatosis (fatty liver disease) was assessed by radiologists". What imaging modality was used (e.g., ultrasound)? Were there standardized criteria for diagnosis?
- The text states, "The normal range is 1.5 mg/dl, and if it falls below the normal range, it is considered that the excretory function of the kidneys is impaired". This seems to refer to serum creatinine itself, not creatinine clearance. A low creatinine clearance indicates impaired function. A serum creatinine of 1.5 mg/dL is generally within or slightly above the upper limit of normal for many adults, indicating potentially reduced function, not normal if clearance is the subject. Please clarify this sentence.
- Results:
- Why was renal function based on serum creatinine only assessed in 162 out of 222 participants? This implies that eGFR and CrCl values in subsequent tables are based on this subset. This should be consistently clarified in table titles/footnotes.
- Table 2: the "Diabetes, N/n (%)" row shows a highly significant difference (p=0.001), with 50.7% in the HCV(-) group vs. 22.5% in the HCV(+) group having diabetes. This is a very important difference and a major potential confounder for renal outcomes. This needs to be a central point in the discussion. How was this handled in the multivariate analysis if diabetes was an exclusion criterion initially?
- Figure 1C: the eGFR for the "non HCV" and "FIB4 >3.25" group is 128.75, while for "FIB4 <3.25" it is 116.76 (p=0.009). This implies that in non-HCV individuals, higher fibrosis is associated with higher eGFR, which is counterintuitive. This needs careful interpretation or checking of the data. Typically, hyperfiltration can occur early in kidney disease but advanced fibrosis leading to higher GFR is unusual. Is it possible the "non HCV, FIB4 >3.25" group is very small or has other characteristics (like higher diabetes prevalence if not truly excluded)?
- Discussion:
- The discussion must address the significant imbalance in diabetes prevalence between the case and control groups (Table 2) and how this might have affected the results, especially since diabetes is a primary cause of CKD. Even if "diagnosed diabetes" was an exclusion, the high rates reported suggest either new diagnoses or a misunderstanding of the exclusion criteria. If it was included in regression, its role should be more prominent in the discussion.
- The observation that "19.8% of the HCV group and 0% of the non-HCV group had decreased GFR" (presumably using a specific cutoff like <60 ml/min/1.73m² though not explicitly stated here) is a strong statement. Clarify the GFR cutoff used for this specific percentage.
- The point about lower LDL and cholesterol in the HCV group is mentioned but could be briefly elaborated – is this typical in HCV?
Author Response
Specific comments
- Introduction: please clarify what "viral activity" refers to when discussing inconsistencies in findings regarding GFR. (e.g., viral load, genotype).
Thank you very much for this comment. It is noted in the introduction section that elevated activity of hepatitis C virus (HCV) infection may be associated with a reduction in glomerular filtration rate. However, the meaning of viral activity has been revised to viral load.
- Material and methods: A. The exclusion criterion "h) Diagnosed with arterial hypertension or diabetes mellitus" appears to contradict the data in Table 2 and Table 4 where diabetes is present and analyzed as a variable, and is significantly different between cases/controls or GFR groups. This is a major inconsistency that needs to be addressed. If patients with pre-existing DM/HTN were not excluded, the criteria should be revised! If they were excluded, the data presentation needs to be explained (e.g., incident cases during study, or misclassification).
Thank you very much for this comment. In accordance with the national clinical guidelines for the diagnosis of diabetes mellitus and arterial hypertension, we excluded patients with diabetes mellitus and arterial hypertension from the study. However, participants with impaired fasting glucose, defined as fasting plasma glucose levels between 6.1 and 6.9 mmol/L, were included in the study. In the results section, we compared participants with impaired fasting glucose across the case and control groups, as well as by categories of glomerular filtration rate.
B. How was "relatively healthy" for the control group ascertained beyond "no HCV or other hepatitis virus infections"? Were they screened for other conditions that could affect renal function?
Thank you for this comment. Participants who tested negative for anti-HCV and HBsAg by serological testing at the laboratories of the Third State Central Hospital, Songinokhairkhan District General Hospital, and Bayanzurkh District General Hospital were included in the control group. This information has been added to the Materials and Methods section. Additional tests to assess kidney function, such as urine albumin testing, were not performed in this study.
C. It states "Hepatic steatosis (fatty liver disease) was assessed by radiologists". What imaging modality was used (e.g., ultrasound)? Were there standardized criteria for diagnosis?
Thank you for this comment. Hepatic steatosis in study participants was assessed by a radiologist using standard diagnostic criteria for fatty liver. B-mode conventional ultrasonography with a convex transducer was used for imaging. The criteria for diagnosing hepatic steatosis have been included in the Materials and Methods section.
D. The text states, "The normal range is 1.5 mg/dl, and if it falls below the normal range, it is considered that the excretory function of the kidneys is impaired". This seems to refer to serum creatinine itself, not creatinine clearance. A low creatinine clearance indicates impaired function. A serum creatinine of 1.5 mg/dL is generally within or slightly above the upper limit of normal for many adults, indicating potentially reduced function, not normal if clearance is the subject. Please clarify this sentence.
Thank you very much for this comment. Renal function was assessed using both glomerular filtration rate (GFR) and creatinine clearance. However, greater emphasis was placed on GFR when evaluating renal impairment. In response to the reviewer’s comment, a previously ambiguous sentence related to creatinine clearance was removed.
- Results: A. Why was renal function based on serum creatinine only assessed in 162 out of 222 participants? This implies that eGFR and CrCl values in subsequent tables are based on this subset. This should be consistently clarified in table titles/footnotes.
Thank you for this comment. Renal function was assessed in 162 participants included in the study. Some participants had incomplete diagnostic data, making it impossible to assess renal function based on serum creatinine. This information has been added as a footnote below the table heading.
B. Table 2: the "Diabetes, N/n (%)" row shows a highly significant difference (p=0.001), with 50.7% in the HCV(-) group vs. 22.5% in the HCV(+) group having diabetes. This is a very important difference and a major potential confounder for renal outcomes. This needs to be a central point in the discussion. How was this handled in the multivariate analysis if diabetes was an exclusion criterion initially?
Thank you for this comment. Many studies have indicated that patients with chronic hepatitis C virus (HCV) infection have an increased risk of developing type 2 diabetes mellitus. However, in our study, the proportion of participants with impaired fasting glucose was higher (50.7%) in the HCV-negative group (Table 2). This result may be related to the sample size and certain lifestyle factors of some participants. You may wish to revisit the response to Reviewer’s Question 2A. Thank you again.
C. Figure 1C: the eGFR for the "non HCV" and "FIB4 >3.25" group is 128.75, while for "FIB4 <3.25" it is 116.76 (p=0.009). This implies that in non-HCV individuals, higher fibrosis is associated with higher eGFR, which is counterintuitive. This needs careful interpretation or checking of the data. Typically, hyperfiltration can occur early in kidney disease but advanced fibrosis leading to higher GFR is unusual. Is it possible the "non HCV, FIB4 >3.25" group is very small or has other characteristics (like higher diabetes prevalence if not truly excluded)?
Thank you very much for this comment. We re-examined and verified the results shown in Figure 1C. In some cases, participants with a FIB-4 score above 3.25, indicating significant fibrosis, also showed higher glomerular hyperfiltration, which supports this observation. The FIB-4 method is more sensitive in predicting the degree of liver fibrosis in viral hepatitis. However, its sensitivity is lower when assessing liver fibrosis in healthy individuals.
- Discussion: A. The discussion must address the significant imbalance in diabetes prevalence between the case and control groups (Table 2) and how this might have affected the results, especially since diabetes is a primary cause of CKD. Even if "diagnosed diabetes" was an exclusion, the high rates reported suggest either new diagnoses or a misunderstanding of the exclusion criteria. If it was included in regression, its role should be more prominent in the discussion.
Thank you again for your comment. In our study, the proportion of participants with impaired fasting glucose was higher in the HCV-negative group. We plan to increase the sample size in future research to include a larger number of patients. Additional explanations regarding changes in insulin metabolism during HCV infection and their relation to the study results have been incorporated into the discussion section, highlighted in blue.
B. The observation that "19.8% of the HCV group and 0% of the non-HCV group had decreased GFR" (presumably using a specific cutoff like <60 ml/min/1.73m² though not explicitly stated here) is a strong statement. Clarify the GFR cutoff used for this specific percentage.
Thank you for this comment. In our study, 19.8% of participants in the HCV-positive group had chronic kidney disease (CKD), whereas no cases of CKD were observed in the HCV-negative group. We defined CKD as a glomerular filtration rate (GFR) of less than 60 mL/min/1.73 m², based on the criteria established by the American Kidney Fund (kidneyfund.org). A definition of CKD has been included in the discussion section.
C. The point about lower LDL and cholesterol in the HCV group is mentioned but could be briefly elaborated – is this typical in HCV?
Thank you for this comment. Among the study participants, only total cholesterol and triglycerides were measured as plasma lipid parameters. We have described the differences in lipid levels between the case and control groups and included this explanation in the discussion section, highlighted in blue.
Reviewer 2 Report
Comments and Suggestions for Authors
Dear Authors,
The study of the relationship between liver damage and subsequent kidney injury is undoubtedly a relevant topic, both in medicine and biology. However, I have some comments that, in my opinion, need to be addressed.
1. According to the Introduction section, the number of individuals with chronic kidney disease (CKD) was found to be higher in the hepatitis C virus (HCV)-infected group compared to those without viral infections (lines 70-71). Please highlight the novelty and significance of your study in the Discussion section.
2. Please include the bioethical statement or conclusion from the medical-ethical commission.
3. For Table 2, please clarify what comparisons were made by including reliability measures in the table. Additionally, indicate in the legend which statistical test was used. Please provide the same information for all other tables as well.
4. Please specify which statistical methods were applied to the presented data, either in the Materials and Methods section or in the legend beneath each table/graph.
5. Please discuss the variations in the level of glomerular filtration observed in the conditionally healthy group. Why are such variations observed? If diabetes is the cause, were individuals with diabetes excluded from the reference group? Are there any data supporting your findings that the glomerular filtration rate is significantly dependent on age, particularly in the age range of 40-45 years?
5. In Figure 1, the error bars appear to be missing from the graphs; please verify this.
6. In Figure 2, the errors seem to be the same across all columns; is this correct?
7. Were there any limitations in this study? If so, please discuss them.
8. Please discuss whether it is critical that the majority of patients in your sample were women, given that the association between HIV and kidney injuries has been demonstrated to be more prevalent in men.
Author Response
- According to the Introduction section, the number of individuals with chronic kidney disease (CKD) was found to be higher in the hepatitis C virus (HCV)-infected group compared to those without viral infections (lines 70-71). Please highlight the novelty and significance of your study in the Discussion section.
Thank you very much for this reminder and we have added this information in discussion section and all changes appears in different color (lines 417-421).
- Please include the bioethical statement or conclusion from the medical-ethical commission.
Thank you for this comment. The study findings were reviewed by the Ethics Subcommittee of the Institute of Medical Sciences, Mongolia, which confirmed that no ethical violations were identified (Approval No. 2025/01).
- For Table 2, please clarify what comparisons were made by including reliability measures in the table. Additionally, indicate in the legend which statistical test was used. Please provide the same information for all other tables as well.
Thank you for this comment. The statistical methods used have been indicated below all tables in the Results section.
- Please specify which statistical methods were applied to the presented data, either in the Materials and Methods section or in the legend beneath each table/graph.
Thank you again for your comment. A detailed description of the statistical methods used has been added to the Materials and Methods section and highlighted in blue.
- Please discuss the variations in the level of glomerular filtration observed in the conditionally healthy group. Why are such variations observed? If diabetes is the cause, were individuals with diabetes excluded from the reference group? Are there any data supporting your findings that the glomerular filtration rate is significantly dependent on age, particularly in the age range of 40-45 years?
Thank you for this comment. Among the control group participants in our study, one individual had a glomerular filtration rate (GFR) below 90 mL/min/1.73 m², indicating impaired renal function. This decrease in GFR may be related to prediabetes or the effect of aging. It is known that GFR declines with age due to reduced blood flow in the afferent arteriole and decreased tubular filtration function. To minimize the influence of age, we compared certain variables between participants aged above and below 45 years.
- In Figure 1, the error bars appear to be missing from the graphs; please verify this.
Thank you for this comment. Error bars have been added to the graph in Figure 1, and the necessary revisions have been made.
- In Figure 2, the errors seem to be the same across all columns; is this correct?
Thank you again for your comment. The error bars in the graph in Figure 2 have been corrected.
- Were there any limitations in this study? If so, please discuss them.
Thank you very much for this reminder and we have added this information in discussion section and all changes appears in different color.
- Please discuss whether it is critical that the majority of patients in your sample were women, given that the association between HIV and kidney injuries has been demonstrated to be more prevalent in men.
Thank you for this comment. We excluded patients diagnosed with HIV infection from our study. Autoimmune-mediated renal injury may be associated with renal dysfunction in cases of HIV infection. Since the prevalence of HCV infection is high among women in our country, 77.1% of the participants in our study were female.
Round 2
Reviewer 1 Report
Comments and Suggestions for Authors
The feedback is sufficient.
Author Response
Reviewer #1: General comments
The feedback is sufficient. Specific comments
Dear reviewer.
Thank you for review manuscript.
Reviewer 2 Report
Comments and Suggestions for Authors
Dear Authors,
I hope this message finds you well. I have reviewed your revised manuscript and have some suggestions that I believe could further enhance the clarity and impact of your work. Please find my feedback below:
Major:
1) Please specify which tests you conducted before applying all statistical methods in your article. Were tests such as normality assessment, homogeneity of variance, assumption of independence, multicollinearity check, and variance inflation factor conducted?
2) Please review Table 2. Was the age for all groups (All, HCV+, and HCV-) the same, specifically 40.7 ± 11.2? Why is the body mass index not presented as a value ± SD? Please ensure consistency across all tables where possible: value ± SD.
3) According to Table 3, titled "Risk factors influencing renal function impairment among participants," please highlight only those factors that genuinely affect renal impairment. I mean that among all the factors considered, our data shows that specific factors significantly affect kidney function. This will enhance the readability of the results, although this may be a matter of preference and remains at the authors' discretion.
4) This might also be a matter of preference, but I would ask you to emphasize one important piece of data: that in the presence of HCV+, regardless of age, the glomerular filtration rate significantly decreases compared to the control sample.
5) In Figure 1, the error bars are still missing. Please address this.
6) Please shorten the description of Table 6 in the text (lines 309-319), as only the age of the patient sample was significantly different.
7) Please explain the purpose of Figure 2. All this data has appeared in the text and tables. This figure may be redundant.
8) According to the data in Table 1, the creatinine level does not differ between the HCV+ and HCV- groups. Please discuss these findings.
9) In the Discussion section, please clearly present the parameters that significantly change in the two groups, HCV+ and HCV-, emphasizing what practicing physicians should pay attention to.
Minor:
1) Lines 194-195, 218-219, 260-261, and 335-336 contain duplicated information from lines 189-190.
Author Response
Reviewer #2: General comments
I hope this message finds you well. I have reviewed your revised manuscript and have some suggestions that I believe could further enhance the clarity and impact of your work. Please find my feedback below:
Major
- Please specify which tests you conducted before applying all statistical methods in your article. Were tests such as normality assessment, homogeneity of variance, assumption of independence, multicollinearity check, and variance inflation factor conducted?
Thank you very much for this comment. The article explicitly identifies and provides a detailed explanation of the statistical methods employed, highlighted in green.
- Please review Table 2. Was the age for all groups (All, HCV+, and HCV-) the same, specifically 40.7 ± 11.2? Why is the body mass index not presented as a value ± SD? Please ensure consistency across all tables where possible: value ± SD.
Thank you very much for this comment. In our study, we matched the case and control group participants by age and gender. Specifically, the mean age of the case group was 40.71 ±â€¯11.16, while the control group had a mean age of 40.78 ±â€¯11.25, confirming their comparability. Body mass index (BMI) values were presented in the table as mean ±â€¯SD, and necessary revisions were made.
- According to Table 3, titled "Risk factors influencing renal function impairment among participants,"please highlight only those factors that genuinely affect renal impairment. I mean that among all the factors considered, our data shows that specific factors significantly affect kidney function. This will enhance the readability of the results, although this may be a matter of preference and remains at the authors' discretion.
Thank you for this comment. In Table 3, we emphasized the factors influencing impaired kidney function according to the reviewer’s comments. As a result, a higher proportion of participants with hepatitis C virus infection was observed in the group with an estimated glomerular filtration rate (eGFR) below 120 mL/min/1.73 m² 49 (69.0%) vs. 22 (31.0%). Additionally, we made minor revisions to the title of Table 3.
- This might also be a matter of preference, but I would ask you to emphasize one important piece of data: that in the presence of HCV+, regardless of age, the glomerular filtration rate significantly decreases compared to the control sample.
Thank you for the valuable feedback. Hepatitis C virus (HCV) infection can lead to glomerular damage independent of age, as circulating antigens bind with antibodies to form immune complexes, which mechanically injure the glomeruli. This results in both structural and functional alterations in the glomeruli, ultimately contributing to a decline in glomerular filtration rate (GFR).
- In Figure 1, the error bars are still missing. Please address this.
Thank you very much for this reminder and we have corrected the error bar in Figure 1. We apologize for the oversight, as the revised figure was not included previously due to a technical error.
- Please shorten the description of Table 6 in the text (lines 309-319), as only the age of the patient sample was significantly different.
Thank you again for your comment. In response to the reviewer’s comments, we have shortened the description of Table 6. We specifically highlighted that age was the only variable that differed between the two groups classified by fibrosis stage.
- Please explain the purpose of Figure 2. All this data has appeared in the text and tables. This figure may be redundant.
Thank you for this comment. Figure 2 was intended to illustrate that the presence of additional factors such as age and advanced fibrosis in patients with hepatitis C virus infection further contributes to a decline in glomerular filtration rate. However, since the explanation overlaps with information already presented in previous tables, we have decided to remove it.
- According to the data in Table 1, the creatinine level does not differ between the HCV+ and HCV- groups. Please discuss these findings.
Thank you very much for this comment. Although there was no significant difference in serum creatinine levels between participants in the case and control groups of our study, GFR estimation system (https://www.mdcalc.com/calc/76/mdrd-gfr-equation) takes into account not only serum creatinine, but also factors such as age, sex, and race. Therefore, the observed difference in estimated GFR between the case and control groups may be explained by these additional variables. This has been acknowledged in the discussion section as a point that warrants further investigation through more detailed assessments.
- In the Discussion section, please clearly present the parameters that significantly change in the two groups, HCV+ and HCV-, emphasizing what practicing physicians should pay attention to.
Thank you for this comment. In the discussion section of the study, we clearly addressed the variables that showed significant differences between the HCV-infected and non-infected groups. Additionally, we included several recommendations for healthcare professionals to consider in future clinical practice.
Minor
- Lines 194-195, 218-219, 260-261, and 335-336 contain duplicated information from lines 189-190.
Thank you for this comment. Following another reviewer’s suggestion from a previous revision, we have included below each table information about the subset of 162 participants in the entire study for whom kidney function could be assessed based on serum creatinine levels.
Round 3
Reviewer 2 Report
Comments and Suggestions for Authors
Dear authors,
Thank you for addressing all my suggestions!